# Cell-free chromatin immunoprecipitation to detect molecular pathways in heart transplantation

Moon Kyoo Jang[1], Tovah E Markowitz[2], Temesgen E Andargie[1,3], Zainab Apalara[1], Skyler Kuhn[2], Sean Agbor-Enoh[1,4]

Existing monitoring approaches in heart transplantation lack the sensitivity to provide deep molecular assessments to guide management, or require endomyocardial biopsy, an invasive and blind procedure that lacks the precision to reliably obtain biopsy samples from diseased sites. This study examined plasma cell-free DNA chromatin immunoprecipitation sequencing (cfChIP-seq) as a noninvasive proxy to define molecular gene sets and sources of tissue injury in heart transplant patients. In healthy controls and in heart transplant patients, cfChIP-seq reliably detected housekeeping genes. cfChIP-seq identified differential gene signals of relevant immune and nonimmune molecular pathways that were predominantly down-regulated in immunosuppressed heart transplant patients compared with healthy controls. cfChIP-seq also identified cell-free DNA tissue sources. Compared with healthy controls, heart transplant patients demonstrated greater cell-free DNA from tissue types associated with heart transplant complications, including the heart, hematopoietic cells, lungs, liver, and vascular endothelium. cfChIP-seq may therefore be a reliable approach to profile dynamic assessments of molecular pathways and sources of tissue injury in heart transplant patients.

## Introduction

Heart transplant patients require close monitoring to identify and treat clinical complications. Endomyocardial biopsy (biopsy) remains the goal standard to monitor and diagnose complications such as acute rejection, cytomegalovirus infection. In addition to a high cost and risk of complications, biopsy samples are examined by histopathology, which is limited by low sensitivity and high inter-operator variability (Marboe et al, 2005). Biopsy plus histopathology also does not provide deep molecular assessments, which may guide treatment decisions. Emerging blood-based approaches may address these limitations (Moss et al, 2018; Andargie et al, 2021;

Sadeh et al, 2021; Vorperian et al, 2022). Recently, cell-free DNA (cfDNA) chromatin immunoprecipitation (cfChIP) has been proposed as one of such an approach; being minimally invasive and reliable to define molecular phenotypes (Vad-Nielsen et al, 2020; Trier Maansson et al, 2023) and the tissue types involved in multiple disease states (Sadeh et al, 2021). Such an approach could serve as a significant advancement to monitor allograft health.

Circulating cfDNA, released into the bloodstream after cell death, is attracting great attention as a novel biomarker for early diagnosis and monitoring in a range of disease conditions (Agbor-Enoh et al, 2019; Duvvuri & Lood, 2019; Jackson Chornenki et al, 2019; Zviran et al, 2020; Brusca et al, 2022). Given its short half-life in circulation (Lo et al, 1999) and sensitivity, cfDNA analysis mimics tissue/end-organ injury with great temporal precision. Earlier studies to characterize cfDNA focused on genetic differences and DNA methylation-based epigenetic signatures. In transplantation, donor-derived cfDNA (ddcfDNA), originating from transplanted organs, is a reliable alternative to biopsy. Current approaches use donor–recipient single nucleotide polymorphisms (SNP) to quantitate ddcfDNA as a measure of allograft injury. Although the SNP-based approach is sensitive in detecting allograft rejection and other complications, it lacks specificity to identify acute rejection phenotypes or define the molecular pathways involved to tailor a treatment. CfDNA is histone bound and maintains histone modifications from its tissue of origin (Sadeh et al, 2021). Posttranslational modifications of histones can regulate genomic elements and is often a proxy for gene expression. The associated genes may show cell/tissue specificity, identifying tissue source involved in disease (Sadeh et al, 2021). Thus, cfChIP-seq can delineate different disease processes, annotate disease phenotypes, and identify tissue injury patterns (Sadeh et al, 2021).

In this pilot study, we assessed if cfChIP-seq can be used to identify biologically plausible gene expression in heart transplant patients and healthy controls, as a first step towards developing specific nucleosome-based cfDNA test that can define molecular phenotypes of heart transplant rejection. We further assessed if cfChIP-seq can identify biological plausible sources of tissue injury in heart transplantation.

[1]Genomic Research Alliance for Transplantation (GRAfT) and Laboratory of Applied Precision Omics, National Heart, Lung, and Blood Institute (NHLBI), NIH, Bethesda, MD, USA [2]NIAID Collaborative Bioinformatics Resource, Integrated Data Sciences Section, National Institute of Allergy and Infectious Diseases, NIH, Bethesda, MD, USA [3]Department of Biology, Howard University, Washington, DC, USA [4]Department of Medicine, Johns Hopkins University, School of Medicine, Baltimore, MD, USA

Correspondence: sean.agbor-enoh@nih.gov

# Results

This study intends to assess if a cfChIP-seq approach would delineate biological processes in physiological conditions and in heart transplantation. We performed a cross-sectional analysis to identify gene sets and cfDNA tissue sources in plasma samples. Housekeeping gene sets in healthy controls, which are constitutionally expressed to maintain basic cell function, was selected to a representative set of genes in physiological conditions. The study also performed differential gene signals between immunosuppressed heart transplant patients and healthy controls to investigate if cfChIP-seq identifies biological plausible pathways in heart transplantation. After consent, blood samples were collected from six healthy controls and nine heart transplant patients (Table S1) for cfChIP-seq. Blood samples were collected in Streck tubes, a prototype sample collection tube containing a proprietary preservative that prevents cell lysis. Plasma was separated from blood cells within 2 h of blood collection by centrifugation to limit cell lysis and genomic DNA contamination (Fig 1A).

Given the potential of genomic DNA contamination during plasma sample preparation, cfDNA quality for each sample was verified by gel electrophoresis or quantitative PCR assessments before cfChIP-seq. All samples analyzed demonstrated high cfDNA quality without relevant genomic DNA contamination as shown in Fig 1B. First, antibodies directed to specific histone modifications were coupled to magnetic beads and incubated with 1 ml of thawed plasma. After washing and digestion of bound histones using proteinase K, captured cfDNA were indexed using Accel-NGS 2S plus DNA library kit with Unique Dual Indexing (Integrated DNA Technologies). Length distribution of cfDNA libraries demonstrate nucleosomal distribution with mononucleosomal predominance and negligible fragments of >1,000 bp indicative of no genomic DNA contamination (Fig 1B). The cfChIPed DNA libraries were subject to paired-end sequencing on the NovaSeq platform at an average of ~40 million reads per sample. Two controls were included for each sample, nonspecific IgG and input cfDNA.

Of the four histone antibodies, H3K4me1 and H3K4me3 reached saturation at ~30,000 million read pairs per sample; the two other histone antibodies (H3K4me2 and H3K36me3) and input cfDNA did not reach saturation (Figs 1C and S1A). The three H3K4me marks followed very similar global distribution (Fig 1D). Their local distributions matched known patterns, with H3K4me3, for example, found mostly near transcriptional start sites (TSS), H3K4me2 found at enhancers and TSS sites, and H3K4me1 found primarily at enhancers of housekeeping (active) genes (Fig S1B). The number and distribution of the called peaks for each histone were consistent with these patterns. For example, 46% of H3K4me3 peaks overlapped promoters, compared with 15% for H3K4me2 and 18% for H3K4me1 (Table S2). Use of the input control samples increased the number of peaks identified for all histone antibodies (Table S2). This also corrected for a weak binding pattern seen on gene bodies for non-expressed genes for all histones (Fig S1B). For these reasons, input normalization was used for all downstream analyses. Furthermore, H3K4me3 showed a higher fraction of reads in peaks (Fig S2A) but a lower number of reads in peaks (Fig S2B), confirming that most of H3K4me3 reads are enriched in the promotor region

that represents a small part of the entire genome. The signals from each of the H3K4me1, H3K4me2, and H3K4me3 were comparable among healthy controls, indicating the reproducibility of the approach (Fig S2).

## CfChIP-seq reliably detect housekeeping genes

We next assessed if cfChIP-seq gene signals are reflective of gene expression, focusing on H3K4me3, which showed highest and most consistent reads per genomic content (RPGC) signal across samples. Prototype housekeeping genes (GAPDH and TBP) showed peaks matching their promoter location, around the TSS. Similarly, for example, monocyte-specific genes (FCN1 and CSF3R), a major cell type contributing plasma cfDNA, showed peaks matching their gene promoter location. However, the monocyte-specific genes showed lower RPGC compared with housekeeping genes that are constitutionally expressed in all tissues. Non-expressed genes in healthy patients (IL-3 and CSF2) showed no or nonspecific peaks, with baseline levels that are no different from nonspecific IgG (Fig 2A). In total, H3K4me3 cfChIP-seq detected 93% of housekeeping genes (Fig 2B), that is 8,452 of the 9,099 housekeeping genes represented in Table S3. Of the 4,070 non-housekeeping genes detected by H3K4me3 cfChIP-seq, one-third (n = 1,381 genes) were neutrophil and/or monocyte-specific genes; neutrophils and monocytes contribute over one-third of plasma cfDNA (Fig 2B). Because leukocytes contribute over three-quarters of plasma cfDNA in healthy patients (Moss et al, 2018), we determined if H3K4me3 cfChIP-seq gene signals correlate with leukocyte ChIP-seq, and observed a strong correlation (Fig 2C). Taken together, these findings are consistent with the prior report (Sadeh et al, 2021) and indicate that H3K4me3 cfChIP-seq reliably detect gene expression signals that are biologically plausible in healthy controls.

## CfChIP-seq identifies relevant molecular pathways in heart transplant patients

We next analyzed plasma from heart transplant recipients maintained on immunosuppression drugs such as tacrolimus. Tacrolimus was initiated at transplantation for these patients and reached therapeutic levels by 2 wk of transplantation. So, posttransplant blood samples were collected at time points with stable tacrolimus levels from 155–223 d after transplantation. The isolated cfDNA showed an expected nucleosomal pattern and were of good quality free of genomic DNA contamination (Fig S3A). H3K4me3 peaks showed the highest frequency around TSS as expected (Fig S3B). Like for healthy controls, cfChIP-seq reliably detected housekeeping genes in heart transplant patients, detecting 8,453 of the 9,099 housekeeping genes. The remaining 3,266 non-housekeeping genes detected were predominantly associated to monocytes and neutrophils (n = 1,333) (Fig 3A). A principal component analysis showed separation of cfChIP peaks of healthy controls and transplant recipients (Fig 3B).

We employed the Deseq2 version of DiffBind v2 (Ross-Innes et al, 2012) to identify differential peaks between healthy controls and heart transplant recipients. There were 12,182 peaks with an FDR less than 0.05 between the healthy controls and heart transplant

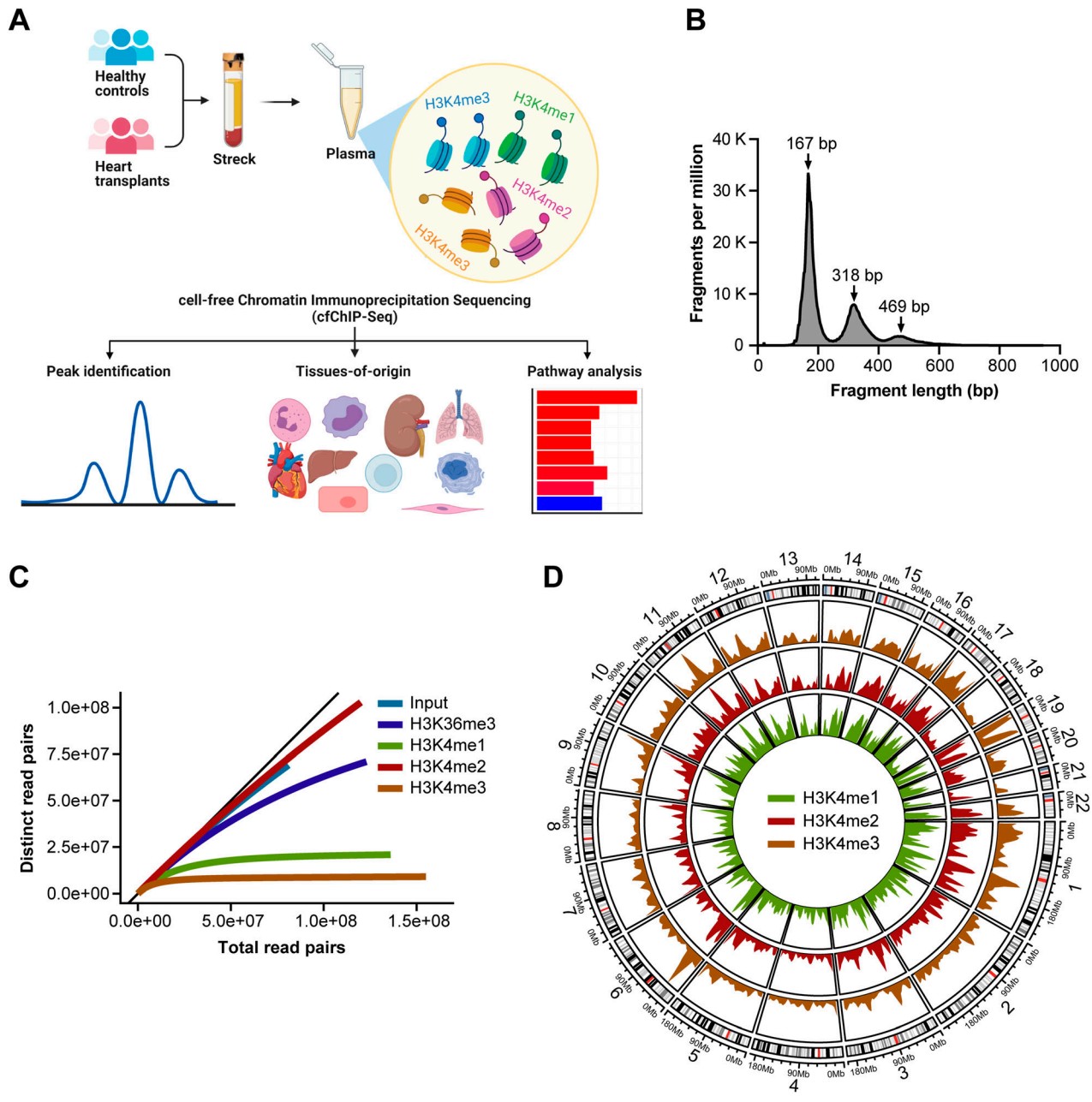

**Figure 1. Design of cell-free chromatin immunoprecipitation sequencing (cfChIP-Seq) of plasma samples.**
**(A)** Schematic workflow: blood samples of healthy controls and heart transplant patients were collected in Streck Cell-Free DNA BCT tubes. After an initial 1,000*g* centrifugation, plasma was further centrifuged at 16,000*g*. Cell-free DNA bound to histones was pooled by specific antibodies which were covalently bound to magnetic beads. Bound cfDNA was purified, used to generate DNA libraries, and sequenced. Sequenced reads were analyzed against controls to identify enrichment peaks, tissue-specific signatures, and biological pathways. **(B)** Length distribution of sequenced DNA fragments by cfChIP-seq showing nucleosomal periodicity. **(C)** Example sequencing saturation curves from HC6 for input DNA, H3K36me3, H3K4me1, H3K4me2, and H3K4me3 reads, highlighting the saturation achieved for H3K4me3 and H3K4me2. **(D)** Chromosome-based circos plots representing the distribution of cfChIP-seq peaks across the genome. Density plot of peaks found in all six healthy controls studied.

samples (Fig 3C). Nearly thirty eight percent (37.9%) of these peaks overlapped the promoter of 4,977 genes. Pathway analysis of these genes identified immune and nonimmune pathways that are relevant to transplantation (Fig 3D and Table S4). The immune pathways were predominantly associated with a loss of H3K4me3 in the promoters in transplant recipients compared with healthy

controls, correlating with the immunosuppressed state of these transplant patients. Of the immune pathways identified, multiple genes showed the differential pattern, including 94 genes in the Rap1 pathway and 82 genes in the Ras signaling pathways. Given that heart transplant patients are maintained on tacrolimus, a calcineurin inhibitor, we further analyzed calcineurin pathway

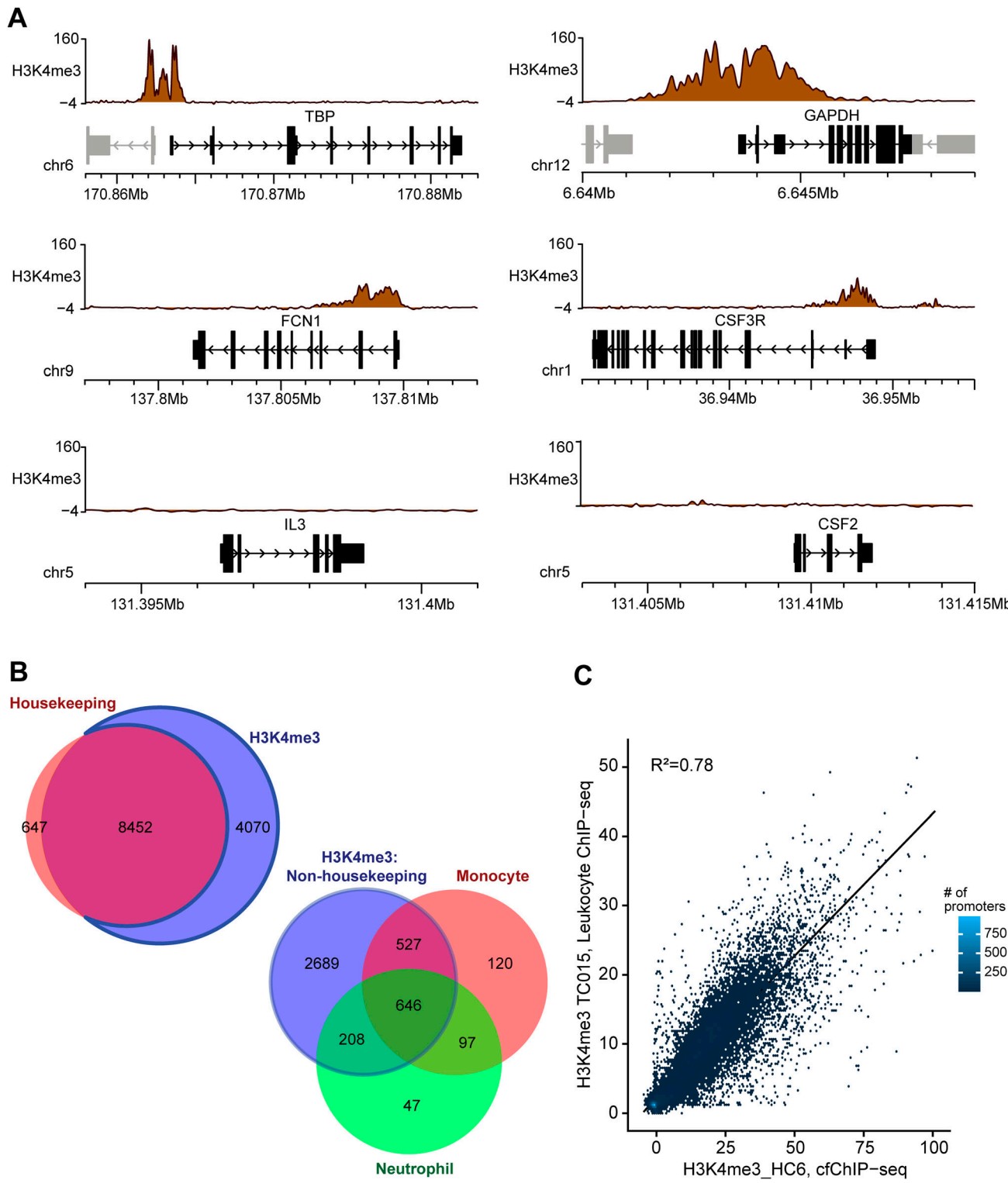

**Figure 2. ChIP-Seq correlates with gene expression in a physiological state.**
**(A)** Epigenome browser snapshots showing (RPGC-normalized sequence reads) H3K4me3 ChIP-seq signals at example housekeeping (TBP, GAPDH), monocyte-specific genes (FCN1, CSF3R) and silent (IL-3, CSF2) genes in HC6. **(B)** Comparison of genes identified by ChIP-Seq. Top: Venn diagram showing the number of promoters overlapping H3K4me3 peaks that are housekeeping genes (red). All gene lists were defined by Sadeh et al (2021) and identified in Table S3. Bottom: Venn diagram showing the overlap of non-housekeeping genes having H3K4me3 peaks associated with monocytes (orange) and/or neutrophils (green). **(C)** Scatterplot plot showing the correlation between H3K4me3 leukocyte ChIP-seq data and cfChIP-seq data ($R^2$ = 0.86). Leukocyte data from Roadmap Epigenomics Consortium et al (2015).

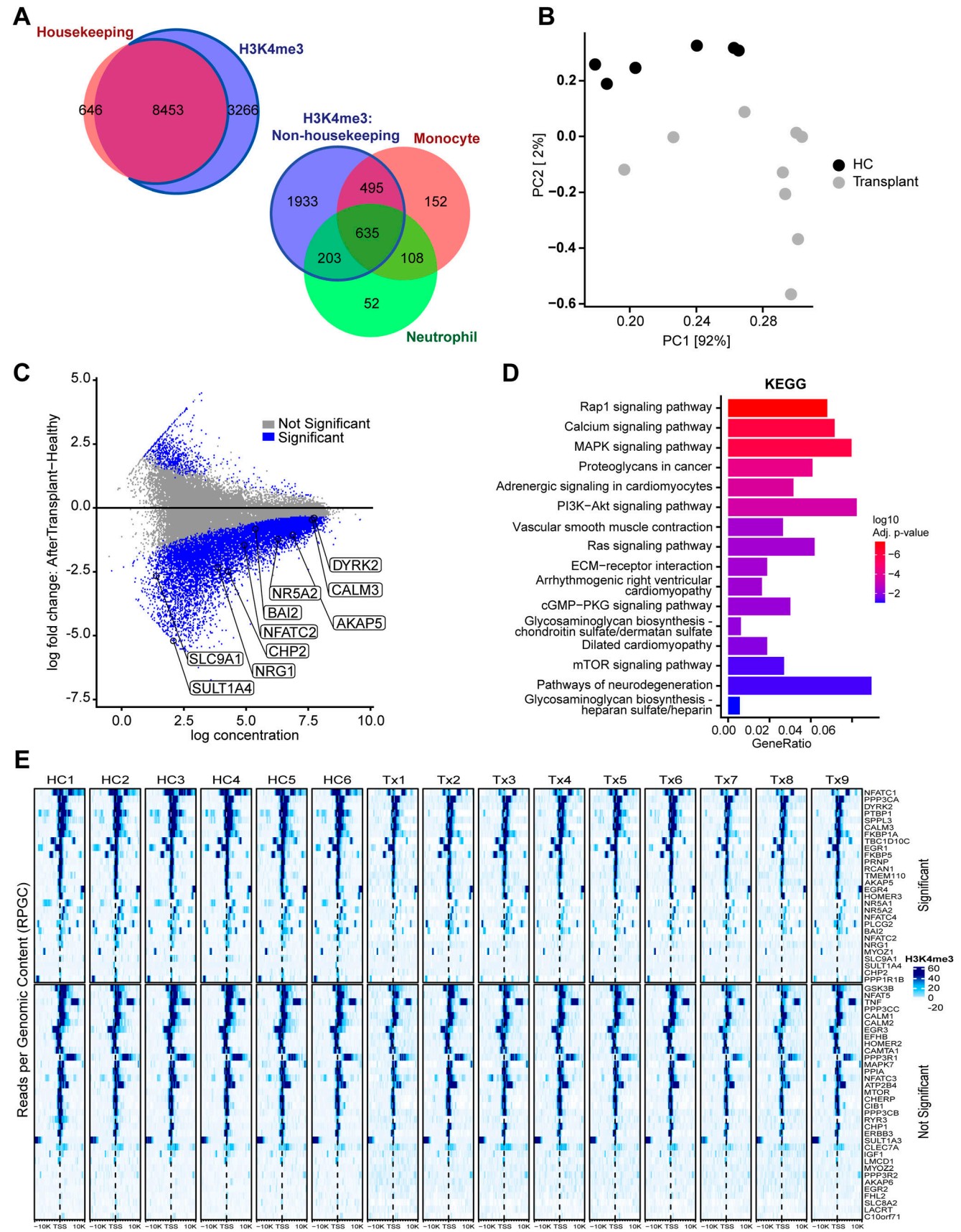

genes. NFACT1, the main target of calcineurin inhibitors, as was 28 of the other 51 calcineurin-related genes showed differential cfChIP-seq signals between heart transplant patients and healthy controls (Fig 3F). 53 genes of the mTOR signaling pathways demonstrated differential peaks, consistent with the fact that these patients were maintained on mTOR inhibitors. Nonimmune pathways also showed differential cfChIP-seq signals (Table S4). Genes associated to reorganization of extracellular matrix and fibrosis showed differences between transplant and healthy controls, including redundant genes associated to extracellular matrix reorganization such as proteoglycans (n = 80) or glycosaminoglycan metabolism (n = 23) (Table S4). Interesting, multiple gene sets associated to neuropathy (n = 136) (Table S4) were also differentially detected in transplant patients; neuropathy because of drug toxicity is a common manifestation in heart transplant patients.

### Different cfDNA tissue sources in transplant patients and healthy controls

CfDNA was assigned to different tissue types using a library of tissue-specific gene signals of 27 tissues (Table S5) (Sadeh et al, 2021). In healthy controls, hematopoietic cells, which include erythroblast, T-cells, B-cells, NK cells, megakaryocytes, monocytes, and neutrophils were major contributors of cfDNA (Fig 4A), contributing 85% of cfDNA (Fig 4B). Non-hematopoietic tissue types, which include skin, adipose, skeletal muscle, pancreas, digestive, liver, brain, lung, vascular tissue, heart, and breast contributed ~15% of cfDNA. The fraction of tissue-specific cfDNA detected by cfChIP-seq correlated with tissue contributions determined using bisulfite sequencing (Fig 4C) using data from a prior publication (Andargie et al, 2021). The latter approach uses tissue-specific DNA methylation signatures to assign tissue-specific cfDNA. Quantitatively, total cfDNA was 5.59 times higher in transplant patients than healthy controls (Fig 4D). Cardiac-specific cfDNA was 2.57 times higher in the heart transplant patients than in healthy controls (Fig 4E), expected in these patients where the heart allograft is exposed to host immunity directed against the allograft. To further assess the greater heart injury in transplant patients, we measured allograft or ddcfDNA by digital droplet PCR assay directed to donor–recipient SNPs. Again, we observed higher levels of ddcfDNA fraction (0.25%) in transplant patients, compared with healthy controls who showed levels similar to the background (<0.01%). In addition, transplant patients demonstrated higher tissue-specific cfChIPed DNA reads from hematopoietic and other non-hematopoietic tissues (Fig 4F and G). For example, liver and vascular endothelial cfDNA were significantly higher in transplant

patients compared with healthy controls, which correlates with the increased frequency of liver and vascular complications in heart transplant patients. Gastrointestinal or neuropathy symptomatology (Díaz et al, 2007; Şahintürk et al, 2021) is observed in transplant patients. We observed correspondingly higher levels of gastrointestinal and neuronal cfDNA in heart transplant patients, which did not reach statistical significance compared with the control probably because of the small sample size. Taken together, cfChIP-seq detects biologically plausible gene expression signals and relevant tissue sources of injury in heart transplant patients.

## Discussion

In heart transplantation, endomyocardial biopsy, the current gold standard does not provide molecular assessments, an important component to provide better phenotyping of transplant complications or guide management strategy. This study demonstrates that plasma cfChIP-seq may be a reliable proxy for gene expression, detecting biologically plausible molecular pathways in heart transplant patients. The approach also identifies sources of tissue injury, identifying injury from tissue types that are associated to transplant complications. The cfChIP-seq approach thus provides a new addition to current monitoring approaches used in heart transplantation, with broader implications in other conditions.

As a proxy for gene expression, cfChIP-seq reliably detected housekeeping genes in healthy control subjects and heart transplant recipients, and differential immune pathways (Fric et al, 2014; Wang et al, 2015; Bonezi et al, 2020) relevant in transplantation. Indeed, several genes of the calcineurin or mTOR pathways were down-regulated in these patients who are maintained on calcineurin and mTOR inhibitors. cfChIP-seq identified differential gene signals related to extracellular matrix reorganization and fibrosis (Chandra et al, 2021; Fan & Hu, 2022). Allograft fibrosis is typically observed 5–15 yr posttransplantation when long-term complications and chronic rejection involving these pathways develop. Additional studies are needed to define the relationship of these early gene changes to chronic allograft fibrosis and chronic rejection, two long-term sequelae that involve extracellular matrix remodeling.

The cfChIP-seq approach also identified tissue contributions of cfDNA that were consistent in a prior report (Sadeh et al, 2021). Heart transplant patients show higher cardiac specific-cfDNA compared with healthy controls, expected in these patients. The higher cardiac injury correlated with high allograft-derived cfDNA. We also observed elevated cfDNA from multiple hematopoietic and

**Figure 3. H3K4me3 cfChIP-seq identifies relevant genes in heart transplant patients.**
**(A)** Venn diagram showing H3K4me3 peaks in transplant patients overlapping promoters of constitutively active housekeeping genes, and non-housekeeping genes associated to monocytes (orange) and neutrophils (green). **(B)** Principal component analysis plot of H3K4me3 peaks showing separation of healthy controls from transplant patient samples. **(C)** MA scatter plot shows differential gene peaks between heart transplant and healthy controls; a subset of calcineurin genes is marked. The blue dots indicate significant differential gene signals between heart transplant and healthy controls, dots under the thick line depict genes with lower H3K4me3 signals in heart transplant subjects compared with healthy controls. **(D)** KEGG pathway enrichment analysis of genes whose promoters were associated with a significant negative fold-change in transplant patients relative to healthy controls. Select immune and nonimmune pathways associated with transplantation state are shown. **(E)** Heatmap showing H3K4me3 pattern around the TSS site of genes associated with the calcineurin pathway. Calcineurin genes were defined as being members of GO: 0097720 or Reactome R-HSA-2025928. Genes above the thick line show significant difference specific to transplantation state as compared with healthy controls.

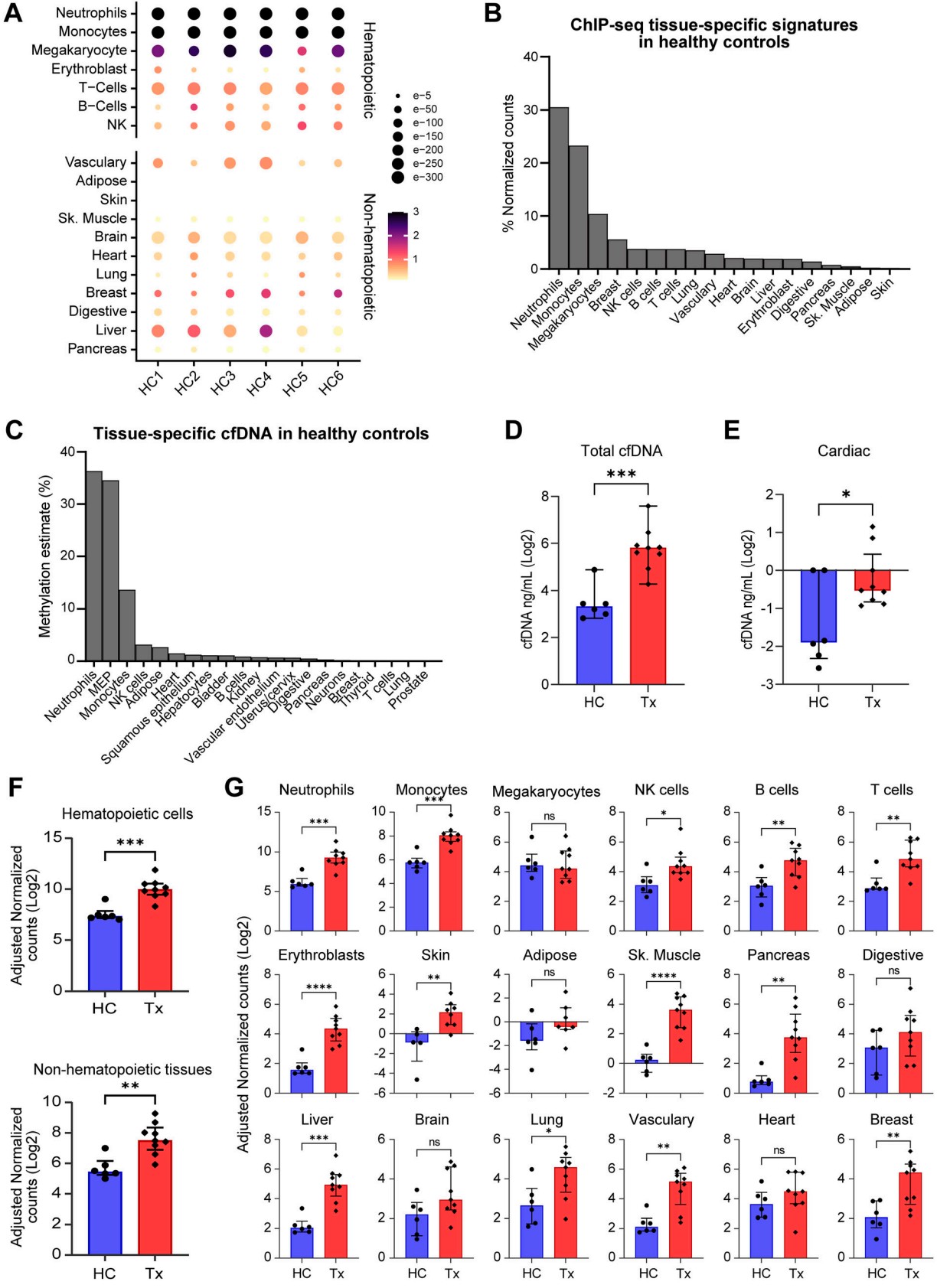

non-hematopoietic tissue types. The tissue–injury pattern observed in heart transplant patients is biologically plausible and represents tissue types with increased incidence of symptomatology observed in these patients because of drug toxicity (Lindenfeld et al, 2004; Tayyem et al, 2018), infections or other complications (Díaz et al, 2007; De Weerdt et al, 2008; Pruitt et al, 2013; Şahintürk et al, 2021).

Although we demonstrate that cfChIP-seq may bring novel insights to advance heart transplant patient monitoring, our study has some limitations. Genomic DNA contamination from cell lysis may occur during blood preparation and would contaminate cfDNA and limit the reliability of cfChIP-seq. To limit this, we used specialized blood collection tubes that prevent cell lysis and process blood samples quickly to separate out cells from plasma. We also demonstrate that, with these steps, genomic DNA contamination is unlikely. Whereas cfChIP-seq identified differential gene expression signals, it lacks the sensitivity to identify the contributions from ddcfDNA. Furthermore, validation of the differential gene signals was not performed. Studies with a larger number of subjects will be relevant to validate this study's findings and further assess the clinical utility. These future studies should sequence cfDNA to higher sequencing depth to fully assess the utility of cfChIP-seq signals for H3K4me1, H3K36me3, and other histone antibodies or to assess the differential gene signals on ddcfDNA. Addition of pre-transplant samples may identify the contribution of pretransplant pathology to posttransplant outcomes. Such studies should also include leukocyte ChIP-seq or functional studies to validate the gene expression signals identified via cfChIP-seq. Pending these additional studies, this pilot study indicates that cfChIP-seq reliably defines gene expression signals that are relevant in heart transplantation. Such a minimally invasive approach may be used to reliably monitor transplant patients and other patient populations.

# Materials and Methods

### Patient recruitment

Healthy adult control plasma samples were obtained at the time of blood donation at the NIH Clinic Center as part of a Department of Transfusion Medicine protocol (99-CC-0168; Collection and Distribution of Blood Component from Healthy Donors for In Vitro Research Use; ClinicalTrials.gov NCT00001846). The protocol uses a questionnaire-based approach to rule out any known chronic disease in healthy controls. Heart transplant patients were recruited as part of the Genome Transplant Dynamics, a multicenter study supported by the Genomic Research Alliance for Transplantation (GRAfT, ClinicalTrials.gov NCT02423070). The study consent and recruit patients awaiting heart transplantation and monitor prospectively after transplant with collection of plasma samples and clinical data. All subjects provided written informed consent.

### cfDNA chromatin immunoprecipitation and library construction

Blood was collected into Cell-Free DNA BCT tube (Streck) and, within 2 h, centrifuged first at 1,600g for 10 min at 4°C, and then at 16,000g for 10 min at 4°C. Aliquots of 1 ml plasma were stored at −80°C until use.

The cfDNA chromatin immunoprecipitation and library construction were performed as described (Sadeh et al, 2021), with minor modifications. In brief, we performed ChIP reactions with covalently conjugated antibodies to magnetic beads to improve the efficiency of ChIPed cfDNA. 50 µg of the antibody were conjugated to 5 mg of epoxy M270 Dynabeads (Invitrogen) according to the manufacturer's instructions. The antibody conjugated beads were stored at 4°C in PBS with 0.02% sodium azide solution. After washing once, 20 µl (0.2 mg, ~2 µg of antibody) of conjugated beads plus 0.1% BSA, 1 ml of plasma was directly added to the rinsed beads in a deep-well plate on magnetic fields. In addition, 1X protease inhibitor cocktail (Roche) and 10 mM EDTA was added, and the reaction was mixed by rotating overnight at 4°C. The beads were magnetized and washed eight times with 150 µl of blood wash buffer (50 mM Tris–HCl, 150 mM NaCl, 1% Triton X-100, 0.1% sodium deoxycholate, 2 mM EDTA, 1× protease inhibitor cocktail) on ice and followed by washing three times with 150 µl of 10 mM Tris pH 7.4 on ice. The histone bound beads were resuspended and the histones were eluted by incubating for 1 h at 55°C in 50 µl of chromatin elution buffer (10 mM Tris pH 8.0, 5 mM EDTA, 300 mM NaCl, 0.6% SDS) containing 50 U of proteinase K (Epicenter). The ChIPed cfDNA were purified by 1.4X SPRI cleanup (AMPure XP, Agencourt) and processed for DNA library construction by indexing on the beads using Accel-NGS 2S plus DNA library kit (IDT).

### Extraction of input cfDNA and NGS sequencing

The volumes of plasma were adjusted to 2.1 ml with PBS containing certain concentration of lambda (λ) DNA shared to ~170 bp to make the final concentration to 0.143 ng/ml plasma. cfDNA was isolated by QIAsymphony circulating DNA kit using a customized 2-ml protocol in 60 µl elution volume. The isolated cfDNA was quantified by qPCR with human Alu115 primers (Forward [F']-CCTGAGGTC

---

**Figure 4. cfChIP-seq identifies tissue sources of cfDNA.**
**(A)** Dot plot showing the cell or tissue contribution of plasma cfDNA in healthy controls estimated. The radius of the circle represents Benjamin–Hochberg-adjusted *P*-values (q-scores) and the color represents the magnitude of normalized reads per kb. **(B)** Bar plot showing the proportion of cfChIP-seq signatures from different tissue types in healthy controls, calculated from the normalized counts. **(C)** Bar plot showing the distribution of plasma cell or tissue-specific cfDNA level for healthy controls using whole-genome bisulfite sequencing (Andargie et al, 2021). **(D)** The concentration of total plasma cfDNA (ng/ml) was determined by qPCR in healthy controls and transplant patients. **(E)** Comparison of cardiac-specific cfDNA in healthy controls and heart transplant patients, measured using cfChIP-seq. Absolute concentration was measured by multiplying cardiac-specific estimate (%) by total cfDNA concentration. **(F)** Comparison of cfChIP-seq hematopoietic and non-hematopoietic tissue signatures in healthy controls and heart transplant patients. **(G)** Comparison of cfChIP-seq tissue-specific signatures in healthy controls and heart transplant patients. For plots (E, F, G), input-normalized reads/kb were multiplied with total cfDNA concentration and comparison was done by Mann–Whitney U test. *P*-value <0.05 was considered statistically significant; *$P < 0.05$; **$P < 0.01$; ***$P < 0.001$.

AGGAGTTCGAG/Reverse [R']-CCCGAGTAGCTGGGATTACA) and Alu247 primers (F'-GTGGCTCACGCCTGTAATC/R'-CAGGCTGGAGTGCAGTGG). The integrity of cfDNA was estimated as the Alu247/Alu115 ratio and the recovery rate calculated with the eluted/added ratio of λ-DNA by qPCR with specific λDNA primers (F'-CGGCGTCAAAAAGAACTTCC/ R'- GCATCCTGAATGCAGCCATA). DNA library was prepared and sequenced on NovaSeq 6000 following standard procedures.

### ChIP-seq processing

Reads were trimmed with Cutadapt version 1.18 (Kechin et al, 2017). All reads aligning to the Encode hg19 v1 blacklist regions (ENCODE Project Consortium et al, 2020) were identified by alignment with BWA version 0.7.17 (Li & Durbin, 2009) and removed with Picard SamToFastq (https://broadinstitute.github.io/picard/). Remaining reads were aligned to an hg19 reference genome using BWA. Preseq 2.0.3 was used to predict read saturation (Daley & Smith, 2013). Reads with a mapQ score less than six were removed with SAMtools version 1.6 (Li et al, 2009) and PCR duplicates were removed with Picard MarkDuplicates. Data were converted into bigwigs for viewing and normalized by RPGC using deepTools version 3.0.1 (Ramírez et al, 2016) using the following parameters: --binSize 25 --smoothLength 75 --effectiveGenomeSize 2700000000 --centerReads --normalizeUsing RPGC. RPGC-normalized input values were subtracted from RPGC-normalized ChIP values of matching cell type genome-wide using deepTools with --binSize 25.

### Peak calling and annotation

Before peak calling, reads aligning to chromosomes X, Y, and M were filtered using SAMtools. Peaks were called using macsNarrow (macs version 2.1.1 from 2016/03/09) (Zhang et al, 2008) with the following parameters: -q 0.01 --keep-dup="all" -f "BAMPE." Differential peaks were called using DiffBind v2 and its Deseq2 differential caller with default parameters (Ross-Innes et al, 2012). Peaks were annotated using UROPA version 4.0.2 (Kondili et al, 2017) and Gencode Release 19 (GrCh37). UROPA annotation conditions involved three query steps, each having an attribute value filter of "protein_coding" and a feature anchor of "start." The three queries varied only by distance which were set as 3,000, 10,000, and 100,000. For gene annotations, only the closest match (finalhit) was used for downstream analyses. For promoter annotations, all genes that matched the first query (allhit) were used for downstream analyses. Over-enrichment analysis was completed on the promoter annotations for H3K4me3 using clusterProfiler (Wu et al, 2021) and plotted with enrichplot. Specifically, the list of genes was compared with the KEGG human pathways with an appropriate background set.

### Statistical analysis and visualization

Graphs were generated by the R software (v3.6.3 or later) using Ggprism, Ggplot2, Ggrepel, Ggpubr, scales, patchwork, RcolorBrewer, VennDiagram, EnrichedHeatmap, circlize, and karyoploteR. Other R packages used were tidyr, dplyr, GenomicRanges, rtracklayer, and rjson. Python software (v3.5) using pybedtools and pysam packages and GraphPad Prism software (v9.4.1) were also used. Comparisons between groups were performed using nonparametric Mann–Whitney U test either on GraphPad Prism or R software (v4.0.3) and adjusted for multiple testing using the Bonferroni correction. $P \leq$ 0.05 indicates statistical significance: $*P < 0.05$, $**P < 0.01$, $***P <$ 0.001, and $****P < 0.0001$.

## Data and Code Availability

The authors declare that all data supporting the findings of this work are available within the article and its supplementary information, and raw sequencing data are available from the corresponding author upon reasonable request. All codes for cfChIP-seq data analysis other than figure generation are available at GitHub repository: https://github.com/OpenOmics/chrom-seek/releases/tag/v1.0.0.

## Supplementary Information

## Acknowledgements

This work was supported, in part, by intramural research funds of the National Heart, Lung, and Blood Institute (NHLBI) (Z99 HL999999) and Lasker Clinical Research Fellowship Program (1-Si2-HL147625). The authors thank Dr. Randall Johnson for assistance in statistical analyses.

### Author Contributions

MK Jang: conceptualization, data curation, formal analysis, visualization, methodology, and writing—original draft, review, and editing.
TE Markowitz: data curation, software, formal analysis, visualization, methodology, and writing—original draft, review, and editing.
TE Andargie: software, formal analysis, methodology, and writing—original draft, review, and editing.
Z Apalara: data curation, software, formal analysis, and writing—original draft, review, and editing.
S Kuhn: data curation, software, methodology, and writing—review and editing.
S Agbor-Enoh: conceptualization, supervision, funding acquisition, investigation, visualization, methodology, project administration, and writing—original draft, review, and editing.

### Conflict of Interest Statement

The authors declare that they have no conflict of interest.

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
