## [Reviewer comments · Life Science Alliance]

Life Science Alliance

Cell-free Chromatin Immunoprecipitation To Detect Molecular Pathways In Heart Transplantation

Moon Kyoo Jang, Tovah Markowitz, Temesgen Andagie, Zainab Apalara, Skyler Kuhn, and Sean Agbor-Enoh
DOI: <https://doi.org/10.26508/lsa.202302003>

Corresponding author(s): Sean Agbor-Enoh, National Institutes of Health

Review Timeline:

Submission Date:	2023-02-20
Editorial Decision:	2023-04-24
Revision Received:	2023-07-21
Editorial Decision:	2023-09-07
Revision Received:	2023-09-11
Accepted:	2023-09-13

Scientific Editor: Novella Guidi

Transaction Report:

April 24, 2023

Re: Life Science Alliance manuscript #LSA-2023-02003-T

Sean Agbor-Enoh
National Heart, Lung, and Blood Institute (NHLBI), NIH

Dear Dr. Agbor-Enoh,

Thank you for submitting your manuscript entitled "Cell-free Chromatin Immunoprecipitation to detect molecular pathways in Physiological and Disease States" to Life Science Alliance. The manuscript was assessed by expert reviewers, whose comments are appended to this letter. We invite you to submit a revised manuscript addressing the Reviewer comments.

Thank you for this interesting contribution to Life Science Alliance. We are looking forward to receiving your revised manuscript.

Sincerely,

B. MANUSCRIPT ORGANIZATION AND FORMATTING:

Reviewer #1 (Comments to the Authors (Required)):

In the manuscript entitled "Cell-free Chromatin Immunoprecipitation to detect molecular pathways in Physiological and Disease States", the authors applied cfChIP-seq with minor modifications to heart transplant patients and healthy controls' plasma. They found cfChIP-seq can reliably detect housekeeping genes in both groups. Next, they identified the pathways were differentially enriched between heart transplant patients and healthy controls. Besides, they also distinguish the tissue of origins of the cfDNA from both groups and that heart transplant patients possess much more cfDNA than healthy control. Besides, they also compare the effects of blood collection methods on the performance of cfChIP-seq. Taken together, the authors showed the potential of cfDNA and cfChIP-seq as a non-invasive strategy to monitor the status of heart transplant patients. Below are some points needs to be clarified.

1. Please check for spelling and grammar. For example, in line 14 Page3, "reproducibility and standardization is important to enable broad applicability", here "is" should be "are".
2. In Fig.3C, the authors showed the cfChIP-seq peaks for H3K4Me3 is highly correlated with those obtained by leukocytes ChIP-seq data, how will you rule out the possibility that cfChIP-seq signal is not from lysis of leukocytes left in the plasma?
3. Could you provide metadata for the healthy samples and heart transplant patients? Like the age, gender, etc
4. Could you also describe the gene signatures obtained from cfDNA origins from non-hematopoietic tissues?
5. In Fig.4D, the authors showed several pathways in which the genes' promoters were differentially enriched in heart transplant patients by cfChIP-seq. Could you indicate if any of the pathways in Fig.4D is uniquely identified by cfChIP-seq but not in ChIP-seq from leukocytes?
6. Since the author also collected plasma samples prior to heart transplant (Page 10, line 2), could you also check the differential enriched pathways before and after heart transplant?
7. In Fig.5, the authors described the tissue specific signatures of cfDNA captured by cfChIP-seq. Could you point out for Fig.5b and Fig.5c, whether healthy control or heart transplant patients's cfDNA or both was used?
8. Please cite the "prior publication" you mentioned on page 11, line 15.
9. Would you be able to show the differentially enriched pathways between healthy control and heart transplant patients by only using the donor derived cfDNA population?

Reviewer #2 (Comments to the Authors (Required)):

The manuscript by Jang et al. describes the use of cfChIP-seq as a non-invasive methodology to proxy gene expression in patients physiological and heart transplant patients taking immunosuppression medications. Overall speaking the manuscript presents novels and interesting information concerning the use of cfChIP-seq as a diagnostic approach based on gene-expression profiles. However, some issues of this generally well-written manuscript need to be resolved. In particular, the methodological results section comparing sample collection in terms of tube types (EDTA and Streck) and plasma centrifugation g's used, is insufficient.

Major concern.

1. The main focus of the manuscript is the description of the use of cfChIP-seq as a diagnostic approach. On top of this, the manuscript is presenting a brief methodological section addressing the use of different types of tubes (EDTA and Streck) and different centrifugation conditions. Methodological optimization can, of course, be beneficial but for the hereby given the derived conclusions appear over-interpreted. Looking at Figure 2, the conclusions that Streck outperforms EDTA (results section p8 lines 7-8) seem largely to be driven by one sample, E3 from individual HC2. Moreover, the conclusion that in particular, S16 (streck with centrifugation g 16) is superior (Page 8 line 16) has an unclear statistical origin. Finally, the number of samples is unequally given S16 for HC3 is excluded making this condition presented with an n=2. To improve this section, the authors need to include more samples to make the conclusions more valid and on top of this present a clear and saturated description of the statistics behind their conclusions including the statistic method used for evaluation for each panel in Figure 2. An alternative suggestion is that the authors simply remove this methodological comparison (figure 2) and the corresponding text throughout the manuscript. As now given the conclusions seem inappropriately substantiated.

Minor concerns.

1. The title should more precisely describe the actual findings in the manuscript related to disease states namely detecting molecular pathways in physiological and heart transplant patients.
2. A brief search of Pubmed shows the presence of additional few references related to cfChIP beyond the key cfChIP-seq reference, Sadeh et al. 2021, which is now given as the single reference to the ChIP methodology in the manuscript. E.g. PMID:

32759018 from 2020 appears to use cfChIP to identify various tissue sources involved in disease (cancer), and despite not using NGS as a result output, this manuscript (and eventually some of the few other cfChIP publications appearing at Pubmed) needs inclusion as reference(s) for the cfChIP methodology (E.g. introduction p4 lines 1-10 and eventual relevant places in the discussion).

3. P7, line 17. Should it be 'for all the analyzed histone modifications'?

4. P8 lines 3 and 4. FRiP abbreviation is given the wrong position.

5. Discussion p13 lines 10 and. The sentence appears erroneous and a reference missing.

6. P16 lines 14 and 15. The text is also given p17 line 4.

Reviewer #3 (Comments to the Authors (Required)):

This manuscript "Cell-free Chromatin Immunoprecipitation to detect molecular pathways in Physiological and Disease States" covers a hot topic, is well executed at the data generation and analysis level but of questionable utility as far as methods and conclusions go.

This is an overall great topic and approaches based on published papers (Sadeh 2021) as well as a review in Science by Dennis Lo.

However, this paper mixes several topics including methods of sample collection and disease entities such as heart transplants and ends up not addressing any of them sufficiently to be of use.

The major weakness is a lack of focus on defined conditions and a low numbers of subjects or samples (sometimes only 2) that make it difficult to assess how much of the findings are random and relevant to the particular subjects picked or the disease or treatment condition or sample collection.

Quantitative and qualitative conclusions (x-fold higher and tissue specificity) as well as testing of different methods of plasma sample origins and preparations are not consolidated by repeat experiments with an independent sample set.

The paper would have been much more useful to the scientific community had the authors followed up on a focused set of samples and analyses.

Reviewer #1 (Comments to the Authors (Required)):

In the manuscript entitled "Cell-free Chromatin Immunoprecipitation to detect molecular pathways in Physiological and Disease States", the authors applied cfChIP-seq with minor modifications to heart transplant patients and healthy controls' plasma. They found cfChIP-seq can reliably detect housekeeping genes in both groups. Next, they identified the pathways were differentially enriched between heart transplant patients and healthy controls. Besides, they also distinguish the tissue of origins of the cfDNA from both groups and that heart transplant patients possess much more cfDNA than healthy control. Besides, they also compare the effects of blood collection methods on the performance of cfChIP-seq. Taken together, the authors showed the potential of cfDNA and cfChIP-seq as a non-invasive strategy to monitor the status of heart transplant patients. Below are some points needs to be clarified.

1. Please check for spelling and grammar. For example, in line 14 Page3, "reproducibility and standardization is important to enable broad applicability", here "is" should be "are".

We thank the reviewer for the careful review. We have performed editorial review throughout the manuscript for spelling and grammar.

2. In Fig.3C, the authors showed the cfChIP-seq peaks for H3K4Me3 is highly correlated with those obtained by leukocytes ChIP-seq data, how will you rule out the possibility that cfChIP-seq signal is not from lysis of leukocytes left in the plasma?

The reviewer raised an important point. We agree that blood preparation can lead to cell lysis and contamination of cfDNA. This could significantly limit the reliability of cfDNA analysis. Our on-going multicenter prospective study is aimed to study the diagnostic performance of cfDNA in transplantation. The sample collection and processing protocols include steps to maintain cfDNA quality and limit genomic DNA contamination. These include:

- Use of specialized cfDNA blood collection tubes to prevent cell lysis. The commercially available blood collection tube, Streck Tubes, has proprietary preservative to stabilize cell membrane and prevent cell lysis.
- Processing blood samples quickly to limit cell lysis. Streck tube allows for blood collection and storage before processing for 24 hours or more without compromising cfDNA quality. In our studies, blood samples are processed within 2 hours of collection to separate cells from plasma and thus further limit the probability of cell lysis.
- Verifying cfDNA quality plasma samples prior to cfChIP-seq experiments. Quality assessment relies on differences in length between cfDNA and the contaminating genomic DNA. cfDNA is predominantly mononucleosomes with peak length of ~160 bp. Di- and tri nucleosome fragments make a small fraction of total cfDNA. Contaminating genomic DNA introduces longer DNA fragments of thousands of base pairs. In such a case, the isolated cfDNA would show a greater fraction of DNA with length > 160 bp. Leveraging this principle, we utilized two analytic methods to check cfDNA quality in the replicate of each sample prior to cfChIP-seq:
 - Integrity Score: We used quantitative PCR and primers directed to *Alu* repeats to measure amplicons of length 115 bp (short fragments that capture mononucleosomal cfDNA and other DNA fragment length) and amplicons of length 247 (longer fragments that should capture contaminating genomic DNA and di- and tri nucleosomal cfDNA). The fraction of long/short fragment or integrity score of the samples in this study was lower than 0.2; integrity scores of ≤ 0.3 are considered high quality.

- Gel electrophoresis on high sensitivity Tap station of isolated cfDNA to profile the length distribution of cfDNA. The electropherogram demonstrates characteristic distribution of high quality cfDNA devoid of contamination, Figure S1.

The probability of cfChIP-seq data coming from lysed leukocytes is therefore much lower. We have included this summary in the Results, Discussion, and Methods.

3. Could you provide metadata for the healthy samples and heart transplant patients? Like the age, gender, etc

Basic demographic data was added to Supplemental Table 1.

4. Could you also describe the gene signatures obtained from cfDNA origins from non-hematopoietic tissues?

Thank you for the question. To clarify, we defined hematopoietic signatures as those associated with the blood: erythroblast, T-cells, B-cells, NK cells, megakaryocytes, monocytes, and neutrophils. Non-hematopoietic signatures are associated with the other unique cell types explored by the NIH Roadmap Epigenomics Consortium and developed by Nature Biotechnology paper in 2021 from Sadeh et al. There are skin, adipose, skeletal muscle, pancreas, digestive, liver, brain, lung, vascular tissue, heart, and breast. We made this clearer in the paper (Result Section, Page 10 Lines 11 – 15). We also created Supplemental Table 5 extracted from the Sadeh et al files showing the coordinates of the windows used for this analysis and the genes they associated with each one.

5. In Fig.4D, the authors showed several pathways in which the genes' promoters were differentially enriched in heart transplant patients by cfChIP-seq. Could you indicate if any of the pathways in Fig.4D is uniquely identified by cfChIP-seq but not in ChIP-seq from leukocytes?

Thank you for this important question. Ideally, we would repeat the computational analyses between healthy individuals and transplant patients with leukocyte ChIP-seq data. This would allow us to 1) validate the differential peaks identify by cfChIP-seq, and then 2) Define distinctive peaks identified by cfChIP-seq but not leukocyte ChIP-seq. However, our extensive search of publicly available data did not identify available datasets of leukocyte ChIP-seq in heart transplant patients. We thank the reviewer for this raising this point. We have included this as a limitation of this study. Discussion Section, Page 13 – 14.

6. Since the author also collected plasma samples prior to heart transplant (Page 10, line 2), could you also check the differential enriched pathways before and after heart transplant?

Thank you for this great question. To strengthen the power of our analysis and to address concerns raised by the reviewers, we have now excluded analysis with small number of samples. We have removed the before transplant samples given the small number (n=2). However, this question is of biological interest to us. In future studies, we intend to examine if pre-transplant biological pathways contribute to post-transplant outcomes. In that study, we plan to analyze and compare enough pre- and post-transplant samples. We thank the reviewer for raising this point. We have included this point in the Discussion.

7. In Fig.5, the authors described the tissue specific signatures of cfDNA captured by cfChIP-seq. Could you point out for Fig.5b and Fig.5c, whether healthy control or heart transplant patients's cfDNA or both was used?

Thank you. Only health control samples were included. The relative profile of tissue-specific cfDNA between healthy controls to transplant patients is shown in Figure 4D – G. We have modified the titles of Figures 4A and B to provide clarity.

8. Please cite the "prior publication" you mentioned on page 11, line 15.
Thank you. The reference now cited in text (Andargie et al., 2021)

9. Would you be able to show the differentially enriched pathways between healthy control and heart transplant patients by only using the donor derived cfDNA population?

Thanks for the great question. The current experimental design limits the feasibility of this analysis. The average %ddcfDNA in heart transplant patients is ~0.1%. With an average sequencing read count of 40 million per sample, only a small fraction of donor reads would be sampled, insufficient to power such an analysis. In future studies, we anticipate developing a targeted cfChIP-seq approach to selectively capture donor reads for such an analysis. We included this point in the Discussion of the manuscript.

Reviewer #2 (Comments to the Authors (Required)):

The manuscript by Jang et al. describes the use of cfChIP-seq as a non-invasive methodology to proxy gene expression in patients physiological and heart transplant patients taking immunosuppression medications. Overall speaking the manuscript presents novels and interesting information concerning the use of cfChIP-seq as a diagnostic approach based on gene-expression profiles. However, some issues of this generally well-written manuscript need to be resolved. In particular, the methodological results section comparing sample collection in terms of tube types (EDTA and Streck) and plasma centrifugation g's used, is insufficient.

Major concern.

1. The main focus of the manuscript is the description of the use of cfChIP-seq as a diagnostic approach. On top of this, the manuscript is presenting a brief methodological section addressing the use of different types of tubes (EDTA and Streck) and different centrifugation conditions. Methodological optimization can, of course, be beneficial but for the hereby given the derived conclusions appear over-interpreted. Looking at Figure 2, the conclusions that Streck outperforms EDTA (results section p8 lines 7-8) seem largely to be driven by one sample, E3 from individual HC2. Moreover, the conclusion that in particular, S16 (streck with centrifugation g 16) is superior (Page 8 line 16) has an unclear statistical origin. Finally, the number of samples is unequally given S16 for HC3 is excluded making this condition presented with an n=2. To improve this section, the authors need to include more samples to make the conclusions more valid and on top of this present a clear and saturated description of the statistics behind their conclusions including the statistic method used for evaluation for each panel in Figure 2. An alternative suggestion is that the authors simply remove this methodological comparison (figure 2) and the corresponding text throughout the manuscript. As now given the conclusions seem inappropriately substantiated.

We thank the reviewer for providing this important context. Our work highlights the potential diagnostic benefit of cfChIP-seq, and as such we have shifted focus from presentation and discussion of methodological comparison to exploration of the potential diagnostic value in a heart transplant cohort. We have now excluded the methodological comparisons from the manuscript. Also, as recommended, we have further increased the samples per group to 9 heart transplant samples and 6 healthy control samples. We have also excluded the small number of pre-transplant samples. Finally, we have made editorial changes throughout the manuscript, including the abstract, introduction, results, and discussion sections. We hope the main message from our findings is now more easily understood that cfChIP is a non-invasive

proxy to determine biologically relevant gene expression signals and profile the sources of tissue injury in heart transplant patients.

Minor concerns.

1. The title should more precisely describe the actual findings in the manuscript related to disease states namely detecting molecular pathways in physiological and heart transplant patients.

After considered the reviewer's suggestion, we changed the tile of this manuscript to 'Cell-free Chromatin Immunoprecipitation to detect molecular pathways in Physiological and Heart Transplant Patients.'

2. A brief search of PubMed shows the presence of additional few references related to cfChIP beyond the key cfChIP-seq reference, Sadeh et al. 2021, which is now given as the single reference to the CHIP methodology in the manuscript. E.g. PMID: 32759018 from 2020 appears to use cfChIP to identify various tissue sources involved in disease (cancer), and despite not using NGS as a result output, this manuscript (and eventually some of the few other cfChIP publications appearing at Pubmed) needs inclusion as reference(s) for the cfChIP methodology (E.g. introduction p4 lines 1-10 and eventual relevant places in the discussion).

We thank the reviewer for this comment and for their careful review. We have included two references (Vad-Nielsen et al. 2020; Trier Maansson et al. 2023) in the manuscript.

3. P7, line 17. Should it be 'for all the analyzed histone modifications'? *The test has been revised.*

4. P8 lines 3 and 4. FRiP abbreviation is given the wrong position.
We thank the reviewer. We have corrected this.

5. Discussion p13 lines 10 and. The sentence appears erroneous and a reference missing.
The sentence has been revised.

6. P16 lines 14 and 15. The text is also given p17 line 4. *We have deleted the same notation on Page 17.*

Reviewer #3 (Comments to the Authors (Required)):

This manuscript "Cell-free Chromatin Immunoprecipitation to detect molecular pathways in Physiological and Disease States" covers a hot topic, is well executed at the data generation and analysis level but of questionable utility as far as methods and conclusions go.

This is an overall great topic and approaches based on published papers (Sadeh 2021) as well as a review in Science by Dennis Lo.

However, this paper mixes several topics including methods of sample collection and disease entities such as heart transplants and ends up not addressing any of them sufficiently to be of use.

The major weakness is a lack of focus on defined conditions and a low numbers of subjects or samples (sometimes only 2) that make it difficult to assess how much of the findings are random and relevant to

the particular subjects picked or the disease or treatment condition or sample collection.

Quantitative and qualitative conclusions (x-fold higher and tissue specificity) as well as testing of different methods of plasma sample origins and preparations are not consolidated by repeat experiments with an independent sample set.

The paper would have been much more useful to the scientific community had the authors followed up on a focused set of samples and analyses.

We thank the reviewer for providing this important context. To be consistent with our main finding, we have refocused the manuscript to highlight that cfChIP-seq has potential diagnostic value in heart transplantation, notably that cfChIP-seq is a non-invasive proxy to determine differential gene expression signals and sources of tissue injury. We have increased the number of representative samples in this pilot study and repeated all analyses with the higher sample size. To maintain a focus on our main message, we have eliminated the methodological analysis that compares cfChIP-seq signals for different sample collection tubes and centrifugation speed. We have therefore excluded data and Figures (Previous Figure 2) of this section. Finally, we performed editorial changes throughout the manuscript to clarify the central message the manuscript.

September 7, 2023

RE: Life Science Alliance Manuscript #LSA-2023-02003-TR

Dr. Sean Agbor-Enoh
National Institutes of Health
National, Heart, Lung, and Blood Institute
10 Center Dr, 7D05
Bethesda, MD 21043

Dear Dr. Agbor-Enoh,

Thank you for submitting your revised manuscript entitled "Cell-free Chromatin Immunoprecipitation To Detect Molecular Pathways In Heart Transplantation". We would be happy to publish your paper in Life Science Alliance pending final revisions necessary to meet our formatting guidelines.

- please upload your main and supplementary figures as single files
- Please upload all figure files as individual ones, including the supplementary figure files
- all figure legends should only appear in the main manuscript file; please add your main, supplementary figure, and table legends to the main manuscript text after the references section
- please add a callout for Figure S2A, S2B to your main manuscript text

A. FINAL FILES:

B. MANUSCRIPT ORGANIZATION AND FORMATTING:

Sincerely,

Reviewer #1 (Comments to the Authors (Required)):

The revised manuscript looks good to me.

Reviewer #2 (Comments to the Authors (Required)):

The manuscript by Jang et al describes the use of cell-free chromatin immunoprecipitation to detect molecular pathways in heart transplantation. In the review process the authors have made a considerable effort to address the concerns from the reviewers. For this reviewer all concerns are appropriately addressed. The introduced corrections have made the manuscript more scientifically sound and in a quality which could merit publication.

September 12, 2023

RE: Life Science Alliance Manuscript #LSA-2023-02003-TRR

Dr. Sean Agbor-Enoh
National Institutes of Health
National, Heart, Lung, and Blood Institute
10 Center Dr, 7D05
Bethesda, MD 21043

Dear Dr. Agbor-Enoh,

Thank you for submitting your Methods entitled "Cell-free Chromatin Immunoprecipitation To Detect Molecular Pathways In Heart Transplantation". It is a pleasure to let you know that your manuscript is now accepted for publication in Life Science Alliance. Congratulations on this interesting work.

DISTRIBUTION OF MATERIALS:

Again, congratulations on a very nice paper. I hope you found the review process to be constructive and are pleased with how the manuscript was handled editorially. We look forward to future exciting submissions from your lab.

Sincerely,
